# A computational text analysis of recent African digital health strategies and their attention to vulnerable populations

**Kimsey Zajac** *, **Louisa Peters, Lutz Maria Kolbe**

Chair of Information Management, University of Göttingen, Göttingen, Germany

* kimsey.zajac@uni-goettingen.de

## Abstract

### Background

Digital health offers opportunities to transform healthcare systems and improve access to care, particularly in low- and middle-income countries. However, there is concern that national digital health strategies may inadequately address the needs of vulnerable populations, risking the reinforcement of existing health inequalities.

### Objective

This study investigates the extent to which recent African national digital health strategies consider and prioritize vulnerable populations particularly at risk of digital exclusion (women, children and the elderly).

### Methods

We conducted a computational text analysis of 13 recent national digital health strategy documents from African countries to identify major topical priorities within the strategies and to systematically assess the frequency and context of references to vulnerable populations.

### Results

Most strategies focus on broad goals such as health system strengthening, infrastructure development, and stakeholder coordination. While there are mentions of women and children, these references are often indirect and not accompanied by concrete plans or dedicated actions. References to the elderly are especially rare. Few strategies include specific measures to ensure digital inclusion or equity for these vulnerable populations.

### Conclusions

Recent African digital health strategies currently prioritize systemic and infrastructural development, but the needs of vulnerable populations are often overlooked or

**Data availability statement:** The data and analysis script underlying the results presented in the study are available from https://doi.org/10.6084/m9.figshare.31169590.

**Funding:** The author(s) received no specific funding for this work.

**Competing interests:** The authors have declared that no competing interests exist.

superficially addressed. To prevent the digital divide from widening, national strategies should embed explicit equity targets, actionable plans, and mechanisms for meaningful inclusion and participation of all population groups.

## Introduction

Digital health holds the promise to fundamentally transform global health systems. By improving the accessibility, quality, and efficiency of healthcare services, digital health solutions are often viewed as enablers of universal health coverage and long-term health system resilience [1–4]. However, many digitalization projects fail or yield negative outcomes, which is both particularly common and harmful in low- and middle-income countries (LMICs) [5–7], where limited infrastructure, low digital literacy, and weak governance hinder the effective rollout of digital health initiatives [8,9]. To increase and improve digital health development and implementation, many countries have adopted digital health strategies. Digital health strategies are comprehensive national or organizational plans that guide the use of digital technologies to strengthen health systems [4]. These strategies provide a structured framework for implementing, monitoring, and evaluating national digital health initiatives and their progress. They also highlight the need to recognize that digital health technologies must be understood within the context of their users, clinical settings, and the broader policy environment in which they operate [4,10]. National digital health strategies can provide essential guidance and structure for aligning stakeholders, allocating resources, and ensuring inclusivity. However, these strategies vary widely, particularly across income groups and geographical regions [11].

The World Health Organization's *Global Strategy on Digital Health* emphasizes the importance of inclusive digital transformation to ensure that "no one is left behind" [4]. Particularly, the strategy highlights that digital health is only valuable if it is "accessible and supports equitable and universal access to quality health services" [4]. Yet, this vision of universal health care through digitalization is challenged by persistent disparities in digital access and literacy, commonly referred to as the digital divide [12,13]. This divide encompasses not only unequal access to devices and internet connectivity but also differences in digital literacy, trust in technology, affordability, social support, data inclusion and availability of culturally and linguistically appropriate content [14,15]. While access to basic technology has improved worldwide, the digital divide remains an issue due to differences in people's ability to use it effectively [16]. Patterns of structural social inequality are often reflected in the digital divide, as those most in need of care are frequently the least likely to have access to or engage with digital technologies [17]. Throughout this paper, we refer to groups affected by structural social inequality as 'vulnerable populations.' Digital technologies are rarely designed with vulnerable populations in mind, often resulting in their underrepresentation in data and algorithms and in solutions that are often unsuitable for their specific needs [18–20]. Hence, vulnerable populations are especially at risk of digital exclusion, both globally and within national contexts [8,21,22].

Structural disadvantages are highly context-specific and rarely the result of a single aspect or affiliation with a specific population group. Rather, vulnerability is often intersectional, particularly in LMICs, where gender, age, geography, disability, socioeconomic status, ethnicity, and displacement can intersect to shape experiences of exclusion [23]. This analysis foregrounds women, elderly, and youth not to suggest that they are the only vulnerable populations affected by digital exclusion, but because these groups have been most consistently examined in digital divide research and are frequently referenced in global policy frameworks. Further, these three groups constitute large, demographically definable and well-established target populations, making them an ideal starting point for assessing digital inclusivity in recent policy frameworks. Gender norms or household dynamics, for instance, are shown to restrict women's access to mobile devices or limit their autonomy in using digital services [24]. Similarly, older adults are less likely to engage with or benefit from digital health solutions and are more prone to experiencing difficulties, anxiety or discomfort when using technology [25–27]. Further, extant literature stresses that children and youth must be prioritized in digital health efforts, as early access to digital technologies is crucial for shaping long-term health outcomes and serves as a key indicator of a society's ability to harness digitalization for universal health coverage [28,29].

Particularly in LMICs, the adoption of digital systems is strongly associated with persistent inequality [30]. As technology advances and reliance on digital tools grows, vulnerable populations that cannot access digital solutions risk becoming more marginalized [31]. However, even when vulnerable populations gain access to digital health interventions, they often cannot derive the same level of benefit from these systems as more advantaged groups [30]. If unaddressed, digital health interventions, and thus national digital health strategies aimed at promoting the spread of digital technologies, may inadvertently reinforce existing inequalities rather than alleviate them [32,33].

Among LMICs, the African continent stands out as a region of both urgent need of and great potential for digital health [34]. Many African countries face serious health system challenges ranging from high disease burdens and underfunded care infrastructure to limited workforce capacity [35]. Digital interventions have shown potential to alleviate these challenges (e.g., through e-health kiosk in rural settings or mobile health platforms) [36,37]. At the same time, the African continent has emerged as a vibrant testing ground for digital health innovations, especially in areas like mobile health due to high mobile phone penetration [38,39]. While previous research has explored the content of digital health strategies globally [11], there is limited knowledge about how countries in Africa conceptualize and operationalize digital health in policy documents and whether vulnerable groups are prioritized. Initial insights from an analysis of strategies from ten African countries suggest that national approaches often overlook the specific health needs and digital risks faced by children and youth – despite their particular vulnerability and the transformative potential of digital technologies for their wellbeing [40]. These findings raise broader questions about whether current strategies are sufficiently responsive to vulnerable populations, or whether they risk reinforcing the very structural inequities they aim to address. We pose the following research question: ***How inclusive are recent African national digital health strategies regarding the needs of vulnerable populations?***

By focusing on African strategies published after 2020, our study complements existing global comparisons of digital health strategies [11] and highlights current region-specific dynamics. The findings provide insight into whether recent African digital health policies are aligned with inclusive health goals and identify potential gaps in addressing the digital divide. In doing so, this paper aims to inform policymakers, practitioners, and researchers seeking to support equitable digital health development on and for the African continent.

## Methodology

### Data collection and preprocessing

This paper represents an initial analysis of how recent African national digital health strategies frame topical priorities and the extent to which they consider vulnerable populations. To identify relevant documents, we systematically searched official government websites, particularly those of African Ministries of Health. Additional reliable sources such as the WHO

Regional Office for Africa were also consulted. We only considered documents published by official institutions to ensure the authenticity and reliability of the data. We included strategies irrespective of language, utilizing online translators for non-English documents. A key criterion for inclusion was temporal relevance: we restricted our analysis to digital health strategies published in or after 2020 and that remained valid at the time of data collection (October 2024). This temporal restriction ensures that only the most recent thematic priorities are captured and enhances comparability across countries. Importantly, this cut-off point also corresponds with the publication of the WHO Global Strategy on Digital Health 2020–2025, which marked a significant shift toward more coordinated, system-level approaches to digital health and has explicitly informed national digital health strategy development. Limiting the analysis to strategies published during this period therefore increases conceptual comparability across countries and ensures that the identified priorities are situated within a shared global policy framework. If multiple versions of a country's national strategy existed, only the most recent version was selected for analysis. Our inclusion criteria further only focused on strategies exclusively dedicated to digital health, excluding broader national health strategies or strategies with only a minor section on digital health. A first search yielded digital health strategies from 33 African countries, of which 13 met our inclusion criteria, namely Botswana [41], Cameroun [42], Ethiopia [43], Ghana [44], DR Congo [45], Malawi [46], Marocco [47], Namibia [48], Sierra Leone [49], Uganda [50], Zambia [51], Zanzibar [52] and Zimbabwe [53].

Due to the extent of text data, we opted for computer-assisted text analysis approaches. This does not only make the analysis more efficient, but can also (in theory) reduce bias induced by human coding, since automated procedures follow predefined rules [54]. We systematically extracted the text content of the identified national digital health strategies, ensuring that all relevant documents were obtained in their entirety including text in tables, graphics and the complete Annex. Before analysis, we preprocessed the text data in a two-step process: In the first step, the text data was normalized by removing punctuation, special characters, and unnecessary spaces, as they do not contribute to insights and cannot be processed by the applied methods. The cleaned texts were then converted into lowercase words using JavaScript. The processed data was saved as plain text files for further analysis. In the second step, Python (version 3.11.5) was used to lemmatize the text, reducing words to their root or base form (e.g., "implementation" or "implementing" to "implement") to ensure consistency across word variants. Subsequently, non-meaningful stopwords – common words like "and," "or," and "but" – were removed, as they do not contribute to thematic analysis. Additionally, context-specific words such as "health," "eHealth," and "healthcare" were excluded because they appear so frequently and broadly across all strategies that they do not help differentiate specific thematic content; therefore, their presence does not meaningfully contribute to identifying unique themes or priorities within the documents. Excluding such context-specific terms enables the analysis to concentrate on more specific, content-rich terms that offer greater thematic relevance and insight.

## Data analysis

To address the research question, we conducted a two-step computer-assisted text analysis of national digital health strategies (see online supplementary material for analysis scripts and pre-processed text corpus). First, we employed Latent Dirichlet Allocation (LDA) to gain a deeper understanding of the topical priorities within the selected sample of digital health strategies. LDA is a generative probabilistic model based on a hierarchical Bayesian model employed in natural language processing for collections of discrete data such as text corpora and is specifically designed to identify patterns within text data. It assumes that documents are mixtures of topics and that each topic is a mixture of words [55]. By revealing latent topics in a text corpus, LDA offers insights into the inherent themes in the text, serving as a form of topic modeling, a predominantly unsupervised machine learning technique that extracts latent topics through observed co-occurrences of words, or "tokens" [55].

To identify latent constructs within the strategies, we first constructed a document-term matrix to represent word frequencies across documents. We then applied the LDA algorithm to model the probability distribution of a set of topics, each characterized by a distribution of words, providing a representation of the underlying themes in the digital health

 

strategies. To determine the optimal number of topics for the LDA model, an iterative parameter optimization process was conducted. In each iteration, parameters such as the number of topics, passes, and iterations were adjusted to improve result quality. The quality of the topic modeling results was assessed using the u_mass coherence score and the inter-topic distance. The goal was to achieve a low u_mass coherence score (i.e., a negative value close to zero), indicating high semantic consistency within topics, while also ensuring a high inter-topic distance to maximize thematic distinctiveness between topics.

Multiple models with varying topic counts were tested. After each iteration, the results were visualized using an inter-topic distance map. The aim was to identify potential overlaps or inconsistencies between topics and to address them by adjusting the model and its hyperparameters accordingly. In addition, visualizing the topics enabled the identification of potential patterns or clusters based on the distances between topics within the model. The model with nine topics was found to be optimal (see Fig 1). A coherence score of −0.08538 was achieved, suggesting that the model produced semantically coherent topics. Overall, we observed that a higher coherence score usually corresponded with poorer results in the inter-topic distance map, which supported the consistency and reliability of the quality assessment. To validate results beyond computational metrics, the topics were manually reviewed to ensure thematic clarity and interpretability.

Finally, we manually assigned descriptive labels to each topic in an iterative process, refining them as our understanding of the data evolved. The validation procedure involved two researchers (first and second author) and a research assistant independently reviewing the word stems associated with each topic and proposing preliminary labels. To support contextual interpretation, Python was used to extract representative text passages for each topic to inform further refinement of the labels. Table 1 illustrates an example of this labeling process (see Table A1 in S1 File in the Annex for the full topic labelling table). In the final step, the first and second author collaboratively reviewed both the word stems and the corresponding sample passages to agree on the most accurate and meaningful labels for each topic. Table A2 in S1 File in the Annex shows the labelling process and the evolution of the topic labels.

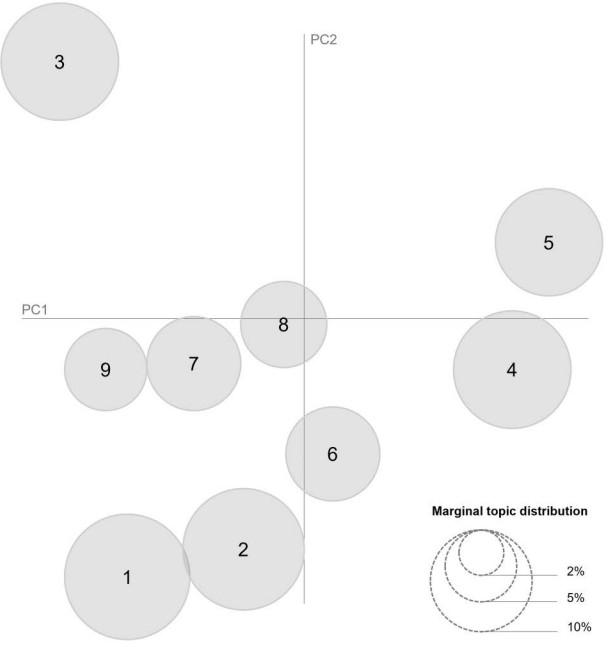

**Fig 1. Inter-topic Distance Map (via multidimensional scaling).**

**Table 1. Topic Labelling Example.**

| Key Terms/Word Stems | Exemplary Strategy Excerpt | Label |
|---|---|---|
| management, resource, level, plan, quality, sector, implementation, care, facility, application, decision, tool, access, infrastructure, development, availability, research, capacity, data, disease | „Successful implementation of the eHealth strategy requires a pool of skilled and competent manpower at all implementation levels. It is therefore critical that a comprehensive eHealth human resource development strategy be developed and implemented to address manpower shortages and technical capacity at central, district, hospital and facility levels." [Namibia] | Digital Health Capacity Building |
| patient, professional, user, care, solution, development, sector, application, term, digitisation, country, telemedicine, hospital, player, security, transformation, exchange, implementation, project | "Systems emerging from the strategy should be designed with the user in mind; and the use of the system should ensure a better experience of the health system, thereby providing more patient-centric, health worker-centric and citizen-centric services." [Namibia] | User-centered Digital Health |

The results from the inter-topical distance map (see Fig 1) indicate a relative clustering of Topics 1, 2, 7, 8, and 9 in the lower left area despite the fact that – apart from Topics 1 and 2 – no direct overlaps between these topics were identified. This observation prompted a more in-depth examination of potential shared thematic features within the text corpus. Thus, based on both the inter-topical distance map and manual higher-order coding based on thematic patterns, we clustered the resulting topics in four overarching priorities.

The second part of our analysis involved the assessment of the inclusion of vulnerable groups in national digital health strategies. To do so, we employed the dictionary approach to quantify references to women, children and the elderly to assess the inclusivity of national digital health strategies. The dictionary-based approach is a widely used method in text analysis that relies on predefined sets of keywords, phrases, or terms associated with specific concepts or categories of interest. These dictionaries enable researchers to systematically scan and categorize text data based on the occurrence of these terms [56]. This method is particularly effective for identifying and extracting segments of text related to targeted themes or topics within large text corpora, allowing for the consistent detection and quantification of concept mentions across documents. To assess the degree of inclusion of vulnerable populations in national digital health strategies, we manually developed a specialized dictionary, which included relevant terms, such as word stems, synonyms, and abbreviations associated with the target population groups, i.e., women and girls, youth, and the elderly. The objective was to capture all relevant mentions related to the population groups of interest. Included terms are based on prior research [11] and expanded by relevant terms used by the WHO to describe these groups. An overview of included terms is presented in Table 2.

Multi-word phrases (e.g., "under five" or "60 and above") included in the dictionary were counted as one mention of the respective vulnerable population. To ensure accuracy and address ambiguous terms, the dictionary results were manually reviewed and false positives were adjusted in the final term counts: For example, if the United Nations International Children's Fund (UNICEF) was mentioned in the strategies, the word "children" would not count as a referral towards youth, since it was only mentioned as the name of a donor organization. To evaluate the prominence of terms related to vulnerable groups within national digital health strategies, we applied normalized weighted term frequencies (NTF). NTF adjusts raw term counts based on individual document length and inverse document frequency to account for terms important to one document but rare across the entire corpus, producing values between 0 and 1. This normalization facilitates comparability across documents of varying lengths. Higher NTF values indicate greater relative importance of a term within a given strategy and across the corpus, while lower values suggest limited relevance or mention.

**Table 2. Overview of Dictionary Terms.**

| Group | Dictionary Key Terms |
|---|---|
| Women | woman, female, women, lady, ladies, girl, girls, mother, mom, gender, feminine, reproductive, maternal, maternity, birth, childbirth, obstetrics, midwifery |
| Youth | child, children, kid, kids, toddler, toddlers, infant, infants, baby, babies, teen, teens, teenager, teenagers, youth, youths, minor, minors, pediatric, juvenile, school age, school aged, under-age, under aged, nursery, elementary, adolescent, under-5, under-five, under five, below 5, below five |
| Elderly | elder, elderly, senior, seniors, aged, ageing, aging, old, older, ancient, veteran, retiree, retired, geriatric, aging population, old age, older age, higher than 60, above 60, 60 and above, higher than 65, above 65, 65 and above, retirement, nursing home, silver generation, dementia, golden years, end of life |

## Results

### Topical priorities of recent African national digital health strategies

Tables 3–6 present an overview of the nine topical priorities identified based on the LDA results derived from recurring token patterns across the text corpus of national digital health strategies. The topics are ordered according to the four overarching priorities based on higher-order coding and the inter-topical distance map. Our LDA analysis shows that recent African national digital health strategies primarily emphasize strengthening health systems through and for the adoption of digital technologies, enhancing health service delivery, establishing the enabling conditions required for digital health to be effective, and promoting comprehensive integration of digital health. Below, we provide explanations for each priority and highlight exemplary quotes from the digital health strategies.

### Health system strengthening

A central theme across African digital health strategies appears to be ongoing systemic issues that make deployment of successful digital health solutions difficult. In particular, challenges such as inadequate infrastructure, limited technical skills, and resource shortages must be addressed to enable effective implementation and sustainability of digital health solutions.

One challenge that is repeatedly highlighted in various strategies is the need to develop digital skills and literacy to build a strong foundation for digital health implementation:

> *"[Existing] challenges are further compounded by the inability of many professionals across the health sector to find sufficient time to up-skill themselves on digital health. … many doctors and nurses believe that meeting ICT requirements is a burden that takes them away from their primary tasks." [Zambia].*

Hence, strong focus lies on building robust and inclusive systems through a combination of technical, institutional, and human-centered approaches. A key priority is enhancing workforce readiness, which includes general population capacity-building, but specifically also health worker training to support digital transformation:

> *"In the most recent training needs assessment, 40% of facilities did not have staff who formally provided IT/systems support. At facilities that did, they lacked the training and mastery levels in key areas of competency for such a role. Addressing staff shortage and equipping them with the skills to provide basic troubleshooting and user support for electronic systems at the facility level will be critical to having a capable workforce." [Malawi].*

A similarly strong focus lies on infrastructure development. All included strategies emphasize the need to expand reliable connectivity, upgrade health facilities, and ensure access to necessary hardware and digital tools*:*

**Table 3. Health System Strengthening.**

| # | Topic Label | Topic Description |
|---|---|---|
| 1 | Digital Health Capacity Building | This topic focuses on preparing health systems by developing the technical skills, human resources, and infrastructure needed for effective digital health implementation. It emphasizes workforce training, resource planning, and creating system readiness to support digital transformation in healthcare. |
| 2 | Health System Reform and Transformation | This topic addresses the comprehensive reform of healthcare systems, combining physical infrastructure development with digital integration. It highlights the need for coordinated policy, cross-sector collaboration, and modernization efforts to ensure equitable, high-quality healthcare delivery. |

**Table 4. Service Delivery.**

| # | Topic Label | Topic Description |
|---|---|---|
| 3 | Digital Health for Targeted Care | This topic highlights how digital tools can enhance care delivery for specific issues, populations or diseases. It focuses on using digital interventions, such as clinical decision support systems for improving health outcomes by streamlining different care forms (e.g., community-based, facility-based) and highlights the importance of integrating technologies along various points of the care journey (i.e., ranging from prevention to treatment). |

**Table 5. Enabling Conditions for Digital Health.**

| # | Topic Label | Topic Description |
|---|---|---|
| 4 | Stakeholder-driven Digital Health Design and Implementation | The topic focuses on how the active involvement of diverse stakeholders – such as government bodies, healthcare providers, technical experts, and communities- benefits the planning, development and rollout of digital health initiatives. It highlights the importance of collaboration, shared ownership, avoiding dual structures and consensus-building in areas like interoperability, standards development, and long-term system integration. |
| 5 | Interoperability and Organizational Alignment | This topic looks at ensuring standardizations, interoperability and aligning health sector organizations around shared digital goals. It emphasizes the importance of coordination between departments, standardized information and communication technology strategies, and cross-functional integration to achieve effective digital health rollout. |
| 6 | Digital Health Governance and Policy | This topic covers the regulatory, legislative, and policy foundations necessary for successful digital health initiatives. It includes governance structures, compliance standards, and leadership roles that ensure accountability and coherence in digital health efforts. |

*"Infrastructure aspects will include at least the following: Workstations, potentially including a wide range of devices such as desktop PCs, mobile tablets and mobile phones, which may be owned and managed by the MOHW or may be the personal devices of health workers or officials." [Botswana].*

Digital health initiatives can only be effective if they are accepted and used. Thus, beyond mere capacity development and access, many strategies also mention the need to overcome resistance and developing a willingness to accept change, particularly among the health workforce:

*"Some of the issues include resistance to use digital solutions; fear of change among health care providers; inadequate basic ICT training and support." [Zanzibar]*

**Table 6. Comprehensive Digital Health Integration.**

| # | Topic Label | Topic Description |
|---|---|---|
| 7 | Strategic Management of Digital Health Interventions | This topic focuses on the planning and management processes that guide digital health development and implementation. It includes risk identification, mitigation approaches, and long-term goal setting to ensure that implementation is both efficient and resilient to disruption. This includes matters of cybersecurity and data privacy. |
| 8 | User-Centered Digital Health | This topic emphasizes the design and development of digital health solutions that prioritize the needs and experiences of both digital health users (e.g., health workers) and beneficiaries (the population) to ensure uptake and value generation. |
| 9 | Coordinated and Integrated Digital Health Implementation | This topic emphasizes the diverse, parallel interventions that are employed to address different health system challenges, ranging from infrastructure development to policy implementation and skill-building, but highlights the importance of coordinated and complimentary approaches. |

## Service delivery

Another key focus of African digital health strategies lies on showcasing the potential of digital interventions to improve healthcare access through enhanced service delivery across the care cascade. This applies both facilitating the general patient pathway as well as improving care for specific diseases or communities.

The strategies reflect an overall optimistic view towards digital health and highlight the importance of integrating it across multiple aspects of patient care:

> *"Digital health should facilitate true continuity of care with holistic, longitudinal patient follow-up." [DR Congo].*

In particular, strategies focus on specific technologies (e.g., point-of-care devices) as well as target populations (e.g., rural communities) and disease areas (e.g., maternal health, infectious and non-communicable diseases) that stand to benefit most from digital solutions:

> *"In the near future, point of care tests that would replace the current MRI, CT scan, and ultrasound would be available not only at referral hospitals but across all levels of health facilities. A typical example is the development of a portable and mobile ultrasound which was unthinkable some years ago." [Ethiopia].*

## Enabling conditions for digital health

The included digital health strategies emphasize the need to build a robust foundation that enables the actualization of expected digital health benefits. Particularly, they stress that successful digital health initiatives depend on inclusive, stakeholder-driven design, strong interoperability and organizational alignment, as well as robust governance and policy frameworks. Without enabling conditions ensuring coordination, accountability, and long-term system integration, digital health cannot achieve the desired effects.

A key topic repeatedly mentioned in many strategies is the necessity of collaboration among governments, healthcare providers, technical experts, and communities to ensure interoperability and avoid isolated initiatives that are not integrated within the broader health system. Further, many strategies also call for coordinated financial investments to spend limited resources wisely:

> *"The success of digital health in Zambia will require relevant government agencies across multiple sectors to collaborate, including health, information and communication technology (ICT), economic, science, innovation, and data*

*privacy and protection agencies. Our government, development partners, and other stakeholders must coordinate investments in the digital health enabling environment to reduce fragmentation of digital health technologies and ensure that different ICT platforms and applications can connect and exchange health information with each other." [Zambia].*

The collaboration among and inclusion of various stakeholders applies across both the design and implementation of digital health initiatives as well as regarding policy development and governance frameworks. Lack of coordination and stakeholder integration can lead to the development of solutions that do not reflect the needs of the users:

*"Some of the issues include […] limited stakeholders' involvement during planning, design and implementation of digital solution, and limited platforms for shared learning to enable stakeholders' access and provide feedbacks." [Zanzibar].*

Stakeholder engagement is often emphasized as a critical prerequisite for developing digital health standards and achieving interoperability, both of which are essential but often insufficiently established in many African countries.

*"An integrated approach to stakeholder management and alignment of prior key initiatives shall be followed. Collaboration between stakeholders is a key criteria to implement digital solutions and services and achieve equitable health coverage. An integrated stakeholder management and engagement platform is necessary for digital technology projects to succeed and for the realization of the vision set out in the blueprint." [Ethiopia].*

Instead, the digital health landscape is usually characterized by silo-solutions that fail to meet the broader health system requirements:

*"Standards and interoperability requirements for digital health solutions are not yet defined, adopted, and implemented. Presently, there is a complex ecosystem of digital health solutions in an undefined, un-integrated and un-exchanged enterprise systems architecture" [Sierra Leone].*

*"A number of challenges were identified such … 61 uncoordinated silo systems in operation." [Namibia].*

This fragmentation is often a result of ineffective regulation. Creating enabling conditions for digital health thus requires strong governance and policy frameworks that provide regulatory guidance, ensure accountability, and enable interoperability:

*"One of the key challenges affecting successful implementation of digital health solutions in the Health Sector is that of weak, uncoordinated and ineffective management frameworks. For the Health Strategy to be effectively implemented, it requires dedicated leadership and governance structures at all levels of implementation." [Botswana].*

**Comprehensive digital health integration**

The selected African digital health strategies further highlight strategic planning and coordinated implementation as essential. They underscore that comprehensive digital health integration requires strategic management, user-centered design, and coordinated, complementary implementation across multiple health system interventions.

Digital health solutions are dynamic rather than one-off implementations and therefore require continuous oversight and adaptation. Many strategies thus underline that successful digital health implementation depends not just on technology and suitable enabling conditions itself, but on strong, adaptive leadership and management practices:

*"For successful implementation of the digital health strategy, leadership and governance shall follow best practices such as digital health initiatives prioritization, periodic monitoring and evaluation, frequent communication between*

*stakeholders, regular review and update of regulations, guidelines, and policies to cope and accommodate technological changes." [Zanzibar].*

Moreover, digital health implementation requires proactive and strategic management rather than reactive problem-solving. Various strategies emphasize the importance of anticipating technical, organizational, and contextual obstacles in advance and putting mitigation strategies in place. Thus, preparedness is highlighted as a key mechanism for reducing both the likelihood of implementation failure and the severity of its potential negative impacts:

*"To avoid failure in the implementation it is important to take a strategic approach to anticipate any potential obstacles for implementation and constitute mitigation strategies. … Preparedness is necessary to mitigate both the probability and the negative consequences." [Malawi].*

Identified risks usually concern limited resources, mismanagement issues, weak coordination or governance mechanisms, resistance to adopt digital health solutions, competing stakeholder interests and security or data privacy threats. Importantly, digital health is often described as fundamentally a socio-technical transformation rather than a purely technological upgrade, aiming to reshape how patients and health workers experience the health system:

*"eHealth is about transforming the user experience, particularly for patients and health workers, so that their experience of the health system changes, care improves, and the health and productivity of all users." [Namibia].*

These benefits of digital health are only achievable when digital health solutions are explicitly designed and implemented around the needs, capacities, and contexts of both users (e.g., health workers) and beneficiaries (e.g., patients). Neglecting these perspectives risks undermining uptake, effective use, and ultimately the intended system-level benefits. Hence, most strategies emphasize user-centeredness, ensuring that digital health solutions are suitable, usable and create the expected values:

*"Developing technology solutions that are user-focused, user-friendly and can adjust to the local context ensures digital health solutions and services consider the local context and focus on the end user." [Ethiopia].*

Finally, recent African digital health strategies often highlight the potential of integrated and adaptive interventions, combining parallel approaches—such as education, regulation, and cybersecurity—to address diverse national health challenges in a flexible and context-sensitive manner:

*"Several emerging digital health interventions can help address the challenges of the health system at different levels, throughout the process leading to universal health coverage." [Cameroon].*

Particularly, they highlight the potential of digital health beyond essentials like hospital information systems, such as community-level interventions:

*"In the digital health era, much can be done beyond in parallel to digitizing health data, depending on the actual need of patients, health care workers, managers, and the community at large." [Ethiopia].*

However, they also highlight the need to coordinate these complementary interventions to ensure they reinforce rather than duplicate each other, maximize impact across different levels of the health system, and maintain alignment with national health priorities and resource capacities. Importantly, this also applies to management of donor funded interventions, that are often uncoordinated, short-term and not adequately integrated into existing systems:

 

*"By 2020, there were over 50 digital health innovations in Uganda by almost as many donors, a situation that has been referred to as "Pilotitis". This tremendous duplication of effort has led to the wastage of scarce resources and resulted in more complicated health systems. Rather than improve health information flow among stakeholders, this series of non-integrated health information systems have created disjointed "information islands" that create barriers to effective communication" [Uganda].*

### Inclusivity of national digital health strategies

Table 7 presents an overview of the dictionary-based analysis results on the inclusivity of recent African national digital health strategies. The analysis of vulnerable populations reveals significant disparities in how frequently these groups are mentioned across different digital health strategies. Specifically, 46.2% of the strategies omit references to children, 30.8% to women, and 61.5% to older adults, with the latter being the most frequently excluded group. The weighted NTF analysis supports these findings, with women showing the highest frequency and presence across the entire text corpus (0.496), followed closely by children (0.446), and the elderly showing a significantly lower NTF and appearance in the text corpus (0.078). Overall, both metrics indicate that women and children receive relatively greater thematic attention compared to the elderly. Eight of the thirteen analyzed countries (61.5%) failed to mention at least one of the vulnerable populations.

The analysis of African national digital health strategies reveals that vulnerable populations *are* referenced across various documents. However, their inclusion tends to be superficial or incidental rather than systematically integrated into digital health planning and implementation. For instance, mentions of vulnerable populations usually appear as part of broader health promotion initiatives rather than through dedicated programs or targeted interventions. There is limited recognition across strategies that certain populations face heightened vulnerability in health contexts. Particularly, women and children are often mentioned in the context of their increased vulnerability to certain diseases and in discussions emphasizing the importance of including them in digital health systems:

*"However, there is a need to heighten the interventions to attain the 95.95.95 targets without leaving children, adolescent girls and young women, among other vulnerable groups, behind." [Zambia].*

However, even when such concerns are raised, they serve primarily to justify general prevention efforts or provide contextual information on population health:

*"Undernutrition is a major problem in the DRC and includes chronic malnutrition, acute malnutrition, and micronutrient deficiencies. Various forms of malnutrition mainly affect young children, pregnant and breastfeeding women, people living with HIV (PLHIV), and the elderly." [DR Congo].*

*"Malaria is the leading cause of morbidity affecting 4 in 10 children aged 6-59 months with an HIV prevalence rate of 1.7%." [Sierra Leone].*

**Table 7. Inclusivity of National Digital Health Strategies.**

| Outcomes | Women | Youth | Elderly |
|---|---|---|---|
| ∑ of mentioned terms | 179 | 162 | 28 |
| % of countries mentioning group | 92.3 | 61.5 | 38.5 |
| NTF | 0.496 | 0.446 | 0.078 |
| Strategies not mentioning group | Cameroun | Cameroun, Ethiopia, Sierra Leone, Uganda, Zambia | Botswana, Cameroun, Ethiopia, Ghana, Zanzibar, Sierra Leone, Uganda, Zambia |

Concrete examples of how digital health can improve these issues are usually missing. The strategies further seldom outline concrete mechanisms to ensure digital inclusion or mitigate the risk of exclusion.

Similarly, references to vulnerable populations are frequently used to describe broader development trends without anchoring them in specific digital health measures. Malawi, for example, highlights closing educational gender gaps and improving women's decision-making power within households as part of its development narrative, without connecting these advances to inclusive digital health initiatives. Other countries, acknowledge persistent health challenges among women and children, despite progress in broader health service delivery metrics:

> *"And although there has been some progress in most of the critical areas of health service delivery and health support systems over the medium term, the health status of most, especially women and children, remains a challenge."* [Zambia].

However, these acknowledgements stop short of translating into concrete digital health interventions aimed at addressing such disparities. Mentions of specific capacity-building or empowerment initiatives for vulnerable populations are rare and scattered. Malawi, for instance, underlines the importance of issuing country-wide digital identities to children to monitor and ensure their inclusion in health systems:

> *"[T]here is need to ensure that all children under the age of 16 are provided with a National Unique ID. Since this population comprises more than 50% of the total population and the most of users of the health system, implementing Unique ID with only adults may not be feasible."* [Malawi].

Yet, such examples remain isolated. Occasionally, strategies do highlight the potential benefits of digital tools for vulnerable populations. Namibia's strategy, for example, recognizes the utility of electronic data in improving maternal and child health services through enhanced tracking and service delivery:

> *"This electronic data […] is also useful for […] maternal and child health services where it can track the individual health problems and treatment over time, giving insights into optimal diagnosis and improving service delivery."* [Namibia].

Other countries, like Zanzibar, mention specific technologies used for tracking reproductive, maternal, newborn or child health services, but do so only as part of long lists of planned or prior interventions. Nonetheless, such acknowledgements are more often descriptive than prescriptive and are not accompanied by dedicated plans or inclusive design approaches.

## Discussion

### Interpretation of results

Overall, the results of our computational text analysis suggest that while vulnerable populations are acknowledged in recent African digital health strategies, their inclusion is not meaningfully operationalized. Recent strategies predominantly emphasize broad digital health goals such as strengthening health systems, increasing resilience and enhancing service delivery and infrastructural development. These topical priorities are aligned with the World Health Organization's Global Strategy on Digital Health [4] and highlight important issues such as workforce skills and general capacity building, interoperability and general barriers like lack of connectivity. These priorities can address the skill and access issues of the digital divide and lay the foundation for increasing the utility of digital health. However, solely focusing on access to and accessibility of digital health interventions overlooks other critical digital divide issues [14]. For everyone to benefit from digital health, strategies should address factors such as (cultural) acceptance, (algorithmic) awareness (i.e., understanding of how automated systems use data and may shape decisions), data inequalities, and the availability of social

and practical support for use [15]. Our findings indicate that some African digital health strategies acknowledge the digital divide and the existence of structural disadvantages for specific population groups. Yet, they tend to lack dedicated frameworks to translate this recognition into equitable access to and benefits from digital health innovations, beyond broad capacity-building measures and vague mentions of vulnerable populations. Therefore, recent African digital health strategies may fall short of the WHO's "leave no one behind" principle for digital health [4].

Through the LDA analysis, we identified four higher-order concepts across the 13 digital health strategies: (1) health system strengthening, (2) service delivery, (3) enabling conditions for digital health and (4) comprehensive digital health integration. Vulnerable populations are not an explicit topical priority. While service delivery and user-centered topics emphasize the importance of considering patients and other beneficiaries, they primarily focus on the general population and do not outline specific inclusion strategies for vulnerable populations. The absence of these groups from the identified topical priorities does not necessarily indicate policy neglect; rather, it suggests that they are not sufficiently prominent in the strategies to emerge as standalone themes. Going beyond the topic modeling analysis, however, the dictionary-based analysis showed that mentions of vulnerable populations within African digital health strategies are rare. Existing references are typically descriptive or incidental rather than constituting substantial programmatic commitments. While the strategies mention women and children more frequently, attention is often superficial, for example, linking these groups to disease burden rather than developing concrete plans for their digital participation. The elderly remain particularly neglected, with little recognition of age-related digital exclusion. While the importance of prioritizing vulnerable populations in digital health efforts is sometimes mentioned, it remains unclear how this inclusion will be achieved. For example, strategies may mention maternal and child health in lists of target areas, but rarely outline mechanisms to ensure that digital interventions genuinely reach and empower these populations.

The tendency for digital health strategies to acknowledge vulnerability in principle while falling short in concrete action points to broader structural dynamics shaping policy formulation. In many African contexts, national digital health planning operates under significant constraints, including limited financial and technical capacity, and competing health system priorities. This difficult foundation can be one reason why strategies often gravitate toward universally applicable, infrastructure-focused goals that are easier to measure and align with donor priorities, such as connectivity, system interoperability, and workforce training. These priorities, while important, tend to absorb policy attention and resources, leaving little room for the more complex, context-sensitive, and long-term investments required to ensure equitable access. Furthermore, the prevailing discourse around digital health frequently frames innovation through a technical lens, reinforcing the idea that solutions are primarily about systems and data rather than people and power. This framing can unintentionally sideline questions of inclusion, making it easier for policymakers to reference vulnerable populations in passing without committing to the structural changes needed to address their specific barriers. Ultimately, the gap between recognition and action is not simply a matter of oversight, but a reflection of how policy is shaped by institutional realities, resource limitations, and the dominant narratives that define what digital health is and who it is for.

Without dedicated accountability mechanisms to ensure that recognition of vulnerability translates into concrete, context-sensitive actions, digital initiatives risk reinforcing existing inequalities [8,31]. The implications are particularly critical in LMICs, where infrastructural constraints and fragmented donor interventions compound risks of pilot-driven, non-integrated, and short-term digital health projects that fail to build inclusive systems [35,50]. To facilitate meaningful inclusion of and capacity-building for vulnerable populations, digital health strategies must move beyond general commitments and embed targeted digital literacy initiatives for vulnerable populations. For instance, future policy development could be strengthened by integrating participatory design approaches that actively involve vulnerable populations, equity impact assessments, and explicit policy frameworks that institutionalize digital equity across the full lifecycle of digital health initiatives. Table 8 demonstrates concrete examples of how language in digital health strategies could be tailored to specifically reflect the needs of each included vulnerable population.

**Table 8. Examples of equity-by-design language for vulnerable populations in policy documents.**

| Women | Digital health platforms for women's health services shall include built-in mechanisms to support health agency in patriarchal or restrictive household contexts, such as optional anonymous access, self-directed appointment scheduling, and secure messaging features that allow women to seek care without requiring spousal or family consent. These platforms will be piloted in partnership with women's rights organizations and local women's groups to ensure that design choices reflect real-world barriers to autonomy, including fear of stigma, surveillance, or retaliation, and will be evaluated for impact on women's ability to initiate and continue care independently. |
|---|---|
| Youth | National digital health platforms for child and adolescent health shall be designed in collaboration with youth advisory councils, schools, and child protection agencies, incorporating age-appropriate language, gamified health education modules, and parental consent pathways that balance privacy with safeguarding. |
| Elderly | Digital health interventions for older adults must be developed through participatory design workshops involving senior community groups, caregivers, and geriatric health professionals, ensuring large-font interfaces, simplified navigation, voice-assisted functionality, and integration with existing community-based health services. Training programs for older adults will be delivered through trusted local institutions (e.g., senior centers and faith-based organizations) and include peer-led support networks to sustain long-term engagement. |

## Limitations and future research

While computational text analysis methods reduce certain forms of bias, human interpretation, especially during the topic labelling of the LDA analysis, remains subjective. Furthermore, while automated content analysis offers insights into the stated priorities of recent African digital health strategies, it is equally important to examine the processes through which they are developed. International bodies such as the WHO emphasize that a participatory approach is essential in formulating such strategies and policies [4,11,40]. While dictionary-based methods are useful to understand large chunks of text data, they remain a shallow proxy and can overlook key issues like structural barriers and intersectional disadvantages. To mitigate this issue, we combined this approach with topic modelling to investigate whether these vulnerable groups are a priority in the included digital health strategies. As discussed above, the LDA analysis did not identify vulnerable populations to be a priority; thus, the dictionary-based approach gives us an indication of the (lack of) relative importance of the studied populations.

However, dictionary-based approaches rely on predefined topics and keywords and are not suited to identify all vulnerable populations, but rather only capture those that we explicitly identified as consistently being referred to across the text corpus. Thus, this study's focus lies solely on women, the elderly and youth. Although these populations are central to discussions on digital inclusivity, we acknowledge that vulnerability is rarely one-dimensional but context-dependent and often intersectional [23,57]. Other vulnerable populations like ethnic minorities, people with disabilities and refugees and multi-dimensional vulnerability remain outside the scope of this paper. They represent vital areas for future qualitative or more granular computational research to better identify and center underrepresented and intersectionally vulnerable populations. Furthermore, although our dictionary was informed by prior research and terminology from global policy frameworks to describe the included vulnerable populations, it may not capture all terms used to describe the three included vulnerable populations. While we conducted manual review of dictionary outputs within the full text of the strategies to identify potential omissions, we cannot rule out the possibility that some relevant references to women, the elderly or youth were missed.

The translation of the non-English documents could further pose an issue. Subtle nuances, idiomatic expressions, and culturally specific terms can be lost or altered in translation, potentially affecting the accuracy of word co-occurrence patterns in topic modeling. Similarly, predefined dictionaries may fail to match translated terms if exact wording or context differs from the original language, which can bias frequency counts or thematic classification. Hence, including non-English

documents in the analysis inevitably introduces some degree of noise and semantic distortion. However, excluding these documents would limit the scope of the study and risk overrepresenting English-language sources.

Finally, we acknowledge that the strategies included in this analysis do not represent all African countries. First, not every African country had published a digital health strategy at the time of data collection. Second, the study deliberately focuses on contemporary digital health strategies published in or after 2020. As a result, our analysis excludes pre-2020 strategies and countries without formally published digital health strategies. Consequently, the generalizability of our findings is limited to recent, formally articulated national strategies rather than African digital health policy more broadly. Future research could conduct longitudinal analyses to assess whether the inclusivity of national digital health strategies has increased, decreased, or shifted in focus over time.

## Conclusion

Recent African national digital health strategies focus on system strengthening and broad enablement but rarely translate equity ambitions into concrete, operational measures for vulnerable populations. Our two-step computer-assisted text analysis of digital health strategies from 13 African countries shows that women, children, and especially the elderly are usually inconsistently and superficially addressed, risking to reinforce existing inequalities as digitalization advances and digital health strategies are put into practice. National digital health strategies should thus embed equity-by-design through explicit targets, action plans, dedicated resources and mechanisms for inclusive participation and evaluation. In practice, this may include community-based digital literacy programs delivered through trusted local intermediaries. It may also involve systematically including representatives of vulnerable populations in the co-design of digital health solutions and in strategy development processes. In addition, national digital health governance frameworks can adopt equity-focused standards, funding criteria, and monitoring indicators.

## Supporting information

**S1 File. Annex.**
(DOCX)

## Author contributions

**Conceptualization:** Kimsey Zajac.

**Data curation:** Kimsey Zajac.

**Formal analysis:** Kimsey Zajac.

**Methodology:** Kimsey Zajac.

**Supervision:** Kimsey Zajac, Lutz Maria Kolbe.

**Validation:** Kimsey Zajac, Louisa Peters.

**Visualization:** Kimsey Zajac.

**Writing – original draft:** Kimsey Zajac, Louisa Peters.

**Writing – review & editing:** Kimsey Zajac, Lutz Maria Kolbe.

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
