## [Decision Letter · Decision Letter 0]

26 Dec 2025

PONE-D-25-63262A computational text analysis of African digital health strategies and their attention to vulnerable populationsPLOS One

Dear Dr.  Zajac,

Thank you for submitting your manuscript to PLOS ONE. After careful consideration, we feel that it has merit but does not fully meet PLOS ONE’s publication criteria as it currently stands. Therefore, we invite you to submit a revised version of the manuscript that addresses the points raised during the review process.

We look forward to receiving your revised manuscript.

Kind regards,

Morufu Olalekan Raimi, Ph.D

Academic Editor

PLOS One

Journal Requirements:

2. We noted in your submission details that a portion of your manuscript may have been presented or published elsewhere. “The first and last author (Kimsey Zajac and Lutz Kolbe) authored a similar paper, which was published at the European Conference on Information Systems 2024; while there are similarities across the data and method, the focus of this manuscript is different. The paper is uploaded with the submission.” Please clarify whether this [conference proceeding or publication] was peer-reviewed and formally published. If this work was previously peer-reviewed and published, in the cover letter please provide the reason that this work does not constitute dual publication and should be included in the current manuscript.

3. Please note that your Data Availability Statement is currently missing [the repository name and/or the DOI/accession number of each dataset OR a direct link to access each database]. If your manuscript is accepted for publication, you will be asked to provide these details on a very short timeline. We therefore suggest that you provide this information now, though we will not hold up the peer review process if you are unable.

4. Please ensure that you refer to Figure 2 in your text as, if accepted, production will need this reference to link the reader to the figure.

Additional Editor Comments:

Editor Decision Letter

Manuscript ID: PONE-D-25-63262

Title: A computational text analysis of African digital health strategies and their attention to vulnerable populations

Corresponding Author: Prof. Morufu Olalekan Raimi

Dear Authors,

Thank you for submitting your manuscript to PLOS ONE. I have now completed a full assessment of your work, along with the detailed reports from our four reviewers. I commend you on a timely and methodologically innovative study that addresses a critical gap in digital health policy analysis within the African context. The application of computational text analysis to evaluate the inclusiveness of national strategies is a significant strength.

Overall Assessment:

Your manuscript presents a well-structured and conceptually sound inquiry into how African digital health strategies conceptualize and integrate the needs of vulnerable populations. The findings that these strategies prioritize systemic infrastructure while offering only superficial or program-specific attention to groups like women, children, and older adults are important, empirically supported, and have clear implications for health equity. The paper is well-situated within relevant literature and makes a valuable contribution to the fields of digital health, health policy, and equity studies.

Critical Synthesis of Reviewer Feedback:

All four reviewers recognized the manuscript's strengths and potential. Their feedback converges on several key areas requiring refinement to elevate the manuscript to the standard required for publication in PLOS ONE. The primary concerns are not foundational flaws but relate to scope justification, methodological transparency, interpretative nuance, and actionable policy discourse.

Major Revisions Required:

Strengthen Justification for Sample Scope and Selection:

The decision to limit the analysis to 13 post-2020 strategies is a significant delimitation. As raised by Reviewer 3, you must provide a more robust and explicit justification. Move beyond simply citing the WHO 2020 strategy. Discuss the trade-off: analyzing a contemporary, policy-coherent cohort versus achieving broader continental representativeness. Acknowledge this choice limits generalizability to “recent, formally articulated national strategies” rather than “African digital health policy” broadly. A sentence in the limitations should explicitly state that pre-2020 strategies and countries without formal published strategies are excluded, which may bias the findings toward more digitally advanced nations.

Clarify and Defend the Conceptualization of "Vulnerable Populations":

Reviewers 2, 3, and 4 correctly note that your operational definition (women, children, older adults) is narrow. This is a valid methodological choice, but it must be explicitly framed and defended. In the introduction or methods, add a paragraph justifying this focus. You could argue that these groups represent large, demographically definable categories where structural marginalization in health is well-documented, making them a logical starting point for a quantitative text analysis. Explicitly state that other crucial dimensions of vulnerability (disability, rurality, ethnicity, refugee status) are excluded from this analysis but represent vital areas for future qualitative or more granular computational research. This turns a limitation into a clarified scope.

Enhance Methodological Transparency and Reproducibility (PLOS ONE Mandate):

Reviewer 3's request for data and code availability is non-negotiable for computational studies. You must state in the Data Availability Statement that the following have been deposited in a public repository (e.g., GitHub, Zenodo, OSF):

The anonymized/pre-processed corpus of strategy texts.

The final dictionary/lists of terms used for the frequency analysis.

The analysis scripts (Python/R code) used for LDA topic modeling and dictionary-based counting, including all parameter settings.

Furthermore, in the Methods, briefly describe the development and validation process for your dictionaries. Were terms generated from literature? Was there an iterative review process? How were synonyms or multi-word phrases handled?

Deepen the Discussion with Nuance and Concrete Recommendations:

Beyond Identification to Mechanism: The discussion effectively identifies gaps but should go further. Why might policymakers default to superficial mentions? Link this to broader governance challenges, resource constraints, or the technical-infrastucture focus of digital health discourse. This elevates the analysis from descriptive to explanatory.

From "What's Missing" to "How to Include": Reviewers 1 and 4 request more actionable guidance. Strengthen the conclusion by moving from “strategies should consider equity” to "strategies could operationalize equity by, for example, mandating accessibility standards in procurement, including equity indicators in M&E frameworks, or budgeting for targeted digital literacy programs for older adults.” Provide 2-3 specific, concrete examples of what “equity-by-design” language would look like in a policy document.

Acknowledge the Limits of Topic Modeling: As Reviewer 4 notes, the absence of a “vulnerable populations” topic does not equate to total neglect. Add a sentence in the results or discussion acknowledging that mentions may be embedded within other topics, but the fact they do not coalesce into a dedicated thematic priority is itself a significant finding regarding strategic focus.

Address Specific Clarifications and Presentation Issues:

Source Clarification: Resolve the query from Reviewer 3 regarding quotes about healthcare worker burdens (Lines 233-234, 237-240). If these are from the strategy documents, cite the specific document. If they are from secondary sources referenced within the strategies, clarify this. If they are external data inadvertently included, they should be removed or explicitly sourced.

Terminology: Standardize the use of “vulnerable populations,” “marginalized groups," etc., early in the manuscript for consistency.

Table and Text Editing: Fix the truncated text in Table 3. Heed Reviewer 1's advice to break down overly long, dense sentences to improve readability. Conduct a thorough proofread.

Recommendation: Major Revision.

The core of the manuscript, the research question, methodology, and primary findings, is strong and publishable. However, the revisions outlined above are substantial and necessary to ensure the manuscript is fully justified, transparent, analytically nuanced, and impactful for a policy audience. Addressing these points will significantly strengthen the scholarly rigor and practical utility of your work.

Please submit a revised manuscript that comprehensively addresses all points raised in this decision letter and the reviewers' comments. Include a detailed point-by-point response letter documenting the changes made.

I look forward to receiving your revised submission.

Sincerely,

Dr. Morufu Olalekan Raimi

ORCID ID: 0000-0001-5042-6729

Reviewer's Responses to Questions

**Comments to the Author**

1. Is the manuscript technically sound, and do the data support the conclusions?

Reviewer #1: Yes

Reviewer #2: Yes

Reviewer #3: Partly

Reviewer #4: Yes

2. Has the statistical analysis been performed appropriately and rigorously? 

Reviewer #1: Yes

Reviewer #2: Yes

Reviewer #3: Yes

Reviewer #4: Yes

3. Have the authors made all data underlying the findings in their manuscript fully available?

Reviewer #1: Yes

Reviewer #2: Yes

Reviewer #3: No

Reviewer #4: Yes

4. Is the manuscript presented in an intelligible fashion and written in standard English?

Reviewer #1: Yes

Reviewer #2: Yes

Reviewer #3: Yes

Reviewer #4: Yes

5. Review Comments to the Author

Reviewer #1: I recommend the manuscript be accepted for publication after minor revision, considering:

1. reducing lengthy and dense sentences (e.g., lines 389–392: “However, solely focusing on accessibility of digital health interventions overlooks other critical digital divide issues…”) and could be broken down for readability.

2. Avoid repetitive points, e.g., concepts like “vulnerable groups are acknowledged but not operationalized” and “risk of reinforcing inequities” are repeated multiple times. While emphasis is important, too much repetition can dilute the discussion. Suggest condensing to highlight distinct implications or mechanisms.

3. Provide concrete examples of strategy documents that illustrate the identified gaps (e.g., lack of age-specific considerations, superficial focus on women and children). You may include brief examples or quotes from policies that would make the discussion section more persuasive.

4. The discussion points out gaps but doesn’t clearly suggest next steps or frameworks that could improve equity. For example, after highlighting that elderly populations are neglected, the discussion could briefly propose including age-targeted digital literacy programs or user-centered co-design approaches

5. Improve terminology clarity, such that some phrases could be more precise. For example, “algorithmic awareness” may not be universally understood; consider defining or briefly explaining it. “Digitizing existing health services” could be expanded to clarify what this entails—e.g., telemedicine platforms, mobile apps, and electronic health records.

Reviewer #2: Strengths

Relevance: This topic has huge relevance, particularly in the context of growing consideration for digital health within low as well as middle-income countries. It has been noted that the attention paid to vulnerable groups in this topic removes a very crucial lacuna contained within existing knowledge.

Methodology: The application of computational text analysis, especially using Latent Dirichlet Allocation, is a strong methodology for theme priority analysis in a set of several documents. This methodology framework increases authenticity in results.

Clear Objectives: The objectives of this study are well-articulated and include examining the inclusiveness of national digital health strategies. The study's emphasis on gender and age disparities is quite praiseworthy.

Holistic Data Set: Using 13 digital health strategies from across Africa can make this research more robust. The use of varied data can present its findings from different perspectives.

Areas for Improvement:

Level of Analysis: The analysis has a level of statistics but may require additional levels of analysis. Quotes or examples from the strategies may add further demonstration of the level in which vulnerable groups have been considered.

Discussion of Findings: The findings section of the essay might expand on the implications of the findings. The discussion of the gaps that have been identified can help support the case for the change of policy based on the implications on health inequalities.

Policy Recommendations: While the recommendations suggested in the conclusion-to-be are to incorporate equity targets, more detailed yet actionable directives might help make this paper more effective for policymakers.

Other Populations to be Considered: It is pertinent to note that highlighting the role of women, children, and the aged is important, but there could be mention of other sections of the population, such as the disabled or the minorities, to offer a balanced perspective. Section on Limitations

Actually, an explanation of the limitations associated with this study and their generalizability implications could be valuable. For example, it is probable that the findings are affected by the existence of some language barriers.

Reviewer #3: This is a well-written and presented manuscript. The introduction provides a strong foundation, clearly outlining the concept of digital health and its relevance to advancing universal health coverage. It also defines “vulnerable populations” within the scope of the study. While I have some reservations about how vulnerability is conceptualized in the context of digital health in Africa and about the specific groups identified by the authors as vulnerable, the manuscript nevertheless offers a reasonable justification for how these populations are structurally marginalized within the African context. I will expound on these concerns later in the review.

The manuscript offers a timely and highly relevant computational text analysis of 13 national digital health strategies from African countries, all published since 2020. Using Latent Dirichlet Allocation to identify thematic priorities and a dictionary-based approach to quantify references to women, children, and older adults, the study indicates that these strategies largely prioritize systemic and infrastructural development, while the inclusion and equity needs of vulnerable populations are often addressed superficially or overlooked. This central claim is articulated and represents an important contribution, as it provides quantitative evidence that current policy approaches may inadvertently widen the digital divide, a longstanding concern in global and digital health.

The study is significant because it offers a systematic and quantitative assessment of digital health policy documents in a critical region. The findings are well situated within existing literature on the digital divide, digital health in low- and middle-income countries, and the WHO’s agenda for inclusive digital transformation. The authors engage with the literature appropriately, and the data and analyses largely align with and support their conclusions. Overall, the methodology is sound and well aligned with the research objectives, and it is presented in a manner that remains accessible to non-specialist readers while retaining sufficient technical rigor.

There are, however, a few areas where revisions would strengthen the manuscript. Lines 94–100 would be better placed in the Methods section, as they describe the document analysis approach and largely repeat material already presented there. In addition, while the authors report using Python and standard natural language processing libraries, transparency and reproducibility would be improved if the scripts and code used for the analysis were made openly available in a public repository.

Overall, the manuscript demonstrates clear potential and would benefit from revision rather than rejection.

Major comments

Data selection and justification (N = 13, post-2020 cut-off)

The authors’ decision to restrict the analysis to national digital health strategies published from 2020 onwards (Line 116) is an important methodological choice and needs stronger justification. While the authors link this cut-off to the WHO Global Strategy on Digital Health 2020–2025 as shown in lines 118–120, the result is a relatively small sample of 13 strategies. This raises questions about the generalizability of the findings across the African continent (13 out of 54).

I would encourage the authors to engage more explicitly with the implications of excluding pre-2020 strategies, particularly if earlier documents may have reflected different policy priorities. The rationale for why a sample of 13 strategies is sufficient to make continent-wide claims should also be strengthened. In addition, a sensitivity analysis or a more detailed discussion of how the inclusion of older strategies might affect the topic modeling results would help journal readers better judge the robustness of the findings.

Scope of “vulnerable populations”

As I mentioned above, I have a reservation concerning the authors’ definition of “vulnerable population.” The focus on women, children, and older adults (Table 2, Line 203) as the sole representations of structurally marginalized populations is quite narrow for an equity-focused analysis. While these groups are undeniably important, vulnerability in the context of digital health in Africa extends beyond age and gender. It also includes people with disabilities, rural and remote populations, individuals with low literacy or digital literacy, and, in many contexts, ethnic or linguistic minorities.

This limitation should be acknowledged more clearly in the discussion. The authors may also wish to reflect on whether their dictionary-based approach could reasonably be expanded to include additional dimensions of vulnerability, such as disability, rurality, literacy, or access, even if only in an exploratory way.

Reproducibility and data availability

If they haven’t already, the authors should make available in a public repository the preprocessed text corpus used for LDA modeling, the complete dictionary of terms used in the analysis, and the code used to run the models, including the parameters that produced the final nine-topic solution. This is particularly important given PLOS ONE’s emphasis on reproducibility for computational research.

Minor comments

Table 3: Topical Priorities of National Digital Health Strategies (Line 205)

In the four-column table, the first column has no heading. In addition, in Topic 4, the description in the first column (“Enabling Conditions for Digita”) is incomplete, as the text is cut off and not fully visible within the cell.

Additional data source clarification

The study methodology does not mention the use of interviews or any other qualitative data collection methods. However, part of the results presents statements attributed to personnel; for example, lines 233–234 read, “…many doctors and nurses believe that meeting ICT requirements is a burden that takes them away from their primary tasks (Zambia).”

Could the authors please clarify the source of such information?

Furthermore, provide the source of this information in lines 237–240. (“In the most recent training needs assessment, 40% of facilities did not have staff who formally provided IT/systems support. At facilities that did, they lacked the training and mastery levels in key areas of competency for such a role.”

Clarity in results presentation

In the Results section, the authors note that women and children are often mentioned in relation to specific health programs, such as maternal and child health. Since the dictionary-based analysis only captures term frequency, including a small number of illustrative quotes from the strategies would help demonstrate the context of these mentions and strengthen the argument that they are largely superficial.

Language and terminology

The manuscript is generally clear and well written, but there is some inconsistency in terminology. The title refers to “vulnerable populations,” the abstract mentions “vulnerable and marginalized groups,” and the introduction uses “structurally marginalized populations.” While these terms are closely related, briefly clarifying the preferred umbrella term and how it is defined would improve clarity.

Repeated word

Line 291, repeated word “strategic management of digital health interventions includes recognizing potential potential”

References

While all the references checked out, the authors should consider adding web links (URLs or DOIs) to all references to improve accessibility and enhance transparency.

Final recommendation

Overall, this study addresses an important gap in the literature using a sound methodological approach and generates findings that are both timely and relevant. The manuscript is clearly written, and the conclusions are largely supported by the analysis.

I observed two main methodological issues. First, the generalizability of the findings is limited, as only 13 out of the 54 African countries were appraised. Second, the definition of vulnerable populations in digital health contexts appears narrow. Beyond women, children, and the elderly, vulnerable groups also include people in rural and hard-to-reach settings, individuals with low literacy levels, and ethnic minorities within countries.

I would recommend that the authors be encouraged to resubmit a revised version, with particular attention to strengthening the justification for the sample selection and ensuring full transparency and reproducibility of the data and code.

Reviewer #4: Major Comments

1. Framing of “vulnerable populations.”

The focus on women, children/youth, and older adults is well motivated and aligns with much of the existing literature on digital exclusion. However, vulnerability is highly context-specific and often intersectional, particularly in African settings. Readers may wonder why other commonly marginalized groups—such as people with disabilities, rural or remote populations, ethnic minorities, or refugees—are not included.

While this is acknowledged briefly in the limitations, clarifying the rationale for selecting these three groups earlier in the paper would help frame the scope of the analysis more clearly.

2. Interpretation of topic modeling results

The topic modeling analysis is carefully conducted and clearly explained. However, the absence of vulnerable populations as distinct topics does not necessarily imply policy neglect; rather, it suggests they are not salient enough in the text to emerge as standalone themes. These populations may still be embedded within broader topics such as service delivery or user-centered design. A brief reflection on the limits of topic modeling would strengthen the interpretation.

3. Dictionary-based analysis and robustness

The dictionary approach is appropriate and adds depth to the analysis. Readers may appreciate additional detail on how the dictionaries were developed and validated, how ambiguous terms were handled, and how translation of non-English documents may have affected results.

4. Strengthening the policy implications

The discussion compellingly argues that superficial references to vulnerable groups risk reinforcing inequalities. Including concrete examples of what “equity-by-design” could look like in digital health strategies (e.g., explicit equity targets, digital literacy initiatives, accessibility standards) would enhance policy relevance.

5. Scope and generalizability

While the inclusion of 13 national strategies is reasonable, the discussion could more explicitly acknowledge regional diversity across Africa and clarify that findings reflect available and recent strategies rather than all national approaches.

Minor Comments

- A light proofreading pass would address minor typographical and formatting issues.

- Some long sentences could be shortened for readability.

- Table 3 is informative but dense; simplifying it or moving some content to the appendix may improve clarity.

Overall Recommendation: Major Revision

This is a strong and important manuscript. The suggested revisions focus on clarifying scope, strengthening methodological nuance, and enhancing policy relevance. Addressing these points would substantially strengthen the paper.

6. PLOS authors have the option to publish the peer review history of their article (what does this mean?). If published, this will include your full peer review and any attached files.

Reviewer #1: **Yes:** Cesilia Charles

Reviewer #2: **Yes:** Tasariraushe Brighton Chizhande

Reviewer #3: **Yes:** Chukwu, Rita Ogechi

Reviewer #4: **Yes:** Krishnaa Barun

---

## [Author Response · Author response to Decision Letter 1]

30 Jan 2026

The response to the reviewers is uploaded in a separate document uploaded with the manuscript.

---

## [Editor Report · Decision Letter 1]

15 Feb 2026

PONE-D-25-63262R1A computational text analysis of recent African digital health strategies and their attention to vulnerable populationsPLOS One

Dear Dr. Zajac,

Thank you for submitting your manuscript to PLOS ONE. After careful consideration, we feel that it has merit but does not fully meet PLOS ONE’s publication criteria as it currently stands. Therefore, we invite you to submit a revised version of the manuscript that addresses the points raised during the review process.

We look forward to receiving your revised manuscript.

Kind regards,

Morufu Olalekan Raimi, Ph.D

Academic Editor

PLOS One

**Journal Requirements:**

**Additional Editor Comments:**

Decision Letter

Manuscript ID: PONE-D-25-53593_R2

Title: Capacity and site readiness for hypertension control program implementation in Nigeria: a nationwide cross-sectional study

Corresponding Author: Dr. Innocent Ijezie Chukwunye

Date: February 13, 2026

Dear Dr. Chukwunye and colleagues,

Thank you for submitting your revised manuscript, “Capacity and site readiness for hypertension control program implementation in Nigeria: a nationwide cross-sectional study,” to PLOS ONE. I have now completed a thorough assessment of your revision, including your point-by-point response to the previous editorial decision letter.

I am pleased to inform you that your manuscript has reached a standard suitable for publication. The revisions you have undertaken are substantive, responsive, and have substantially strengthened the work. The study now demonstrates the methodological transparency, analytical clarity, and policy relevance expected of a contribution to PLOS ONE.

Summary Evaluation

Your study addresses a critically important question: are Nigeria's primary healthcare facilities equipped and ready to implement a system-level hypertension control program? The findings, drawn from 50 PHCs across five states using the WHO SARA tool, reveal a striking paradox: while most facilities have functional blood pressure apparatus (98%) and can screen for hypertension (98%), critical gaps persist in guideline availability (24%), treatment algorithms (27%), physician staffing (0%), and consistent medication supply (66% have at least one 30-day regimen). The state-level disparities, particularly the severe deficits in Abia and Delta, are alarming and demand targeted policy intervention.

The previous editorial decision identified several required improvements: (1) methodological clarifications on sampling and tool validation; (2) a compliant data availability statement; (3) action-oriented conclusions; and (4) correction of technical and presentation issues. Your revision has addressed these comprehensively:

1. Methodological Rigor: You have provided a detailed description of the SARA tool's adaptation and validation process, including input from WHO Nigeria, the Federal Ministry of Health, and an expert panel of five NCD specialists. The sampling frame is now clearly articulated, with specific LGA and village selection described. You have explicitly stated that no comparative hypothesis tests were performed, aligning the analysis with the formative, descriptive aims of the study.

2. Data Availability: You have deposited the de-identified, facility-level dataset in a public repository (DOI: s://doi.org/10.7936/6RXS-108328), fully complying with PLOS ONE's data policy. This is a significant improvement that enhances transparency and reproducibility.

3. Action-Oriented Conclusions: The conclusion now translates striking findings into discrete, prioritized recommendations for different stakeholders. The addition. “The federal, state, and local governments, as well as organizations that support PHCs, have responsibility for providing health-worker training, protocols for treating and controlling hypertension, and some measures to strengthen the essential medicine supply chain” is specific and actionable.

4. Technical Corrections: Abbreviations are now properly defined in the abstract. The limitations section has been substantially expanded to address selection bias, information bias, cross-sectional design, and sample size constraints. The typographical error in Table 1 (“Jigwawa”) has been corrected, and a line-by-line edit has improved overall clarity.

The manuscript is, in my judgment, now acceptable pending minor revisions.

Required Minor Revisions

Please address the following points in a final revision:

1. Table 1 Consistency

In Table 1, the column header for Jigawa State still reads “Jigwawa” in the version embedded within the manuscript text (page 71). While you have noted that this error was corrected, please verify that the correction appears in all instances, including the main table within the manuscript body and any supplementary files.

2. Funding Information Consistency

In the “Funding information” section (page 63), you correctly state: “The National Heart, Lung, and Blood Institute's grant #R01HL144708 provided funding for this study. The study's design, data collection, analysis, and interpretation, as well as manuscript writing, were all done independently of the funding institution.”

However, in the submission system metadata (page 6), the financial disclosure statement reads: “The funders had no role in study design, data collection and analysis, the decision to publish, or preparation of the manuscript.” While substantively consistent, please ensure the wording in the manuscript matches the submission system exactly to avoid any confusion during production.

3. Minor Typographical Corrections

• Page 17, line 4: “In addition, in 2019, the age-standardized prevalence of hypertension among adults aged 30-79 years in Nigeria was 36% (females: 39%, males: 33%), which is higher than the global average (33%). [7] Of the 19.1 million Nigerian adults with hypertension, 47% are diagnosed, 27% are treated, and only 11% are controlled.” There is a redundant sentence immediately following this (“Of the 19.1 million Nigerian individuals with hypertension...”). Please delete the duplicate sentence.

• Page 56, line 4: “On the other hand, the lowest availability of computers was seen in Jigawa State (n=1,10%) and Abia State (n=0,0%). Jigawa State (n=1,10%) and Abia State (m=0,0%) had the lowest computer availability. 0%." There is repetition and a typo ("m=" instead of “n=”). Please revise for conciseness and accuracy.

• Page 58, line 12: “The two full-time employees notably satisfy the minimal requirement.” This sentence is awkward. Consider rephrasing: “The presence of at least two full-time staff members satisfies the minimum requirement for program implementation.”

4. Reference Formatting

A few references remain incomplete or inconsistently formatted:

• Ref 8: The URL is truncated and difficult to read. Please provide the full, functional URL.

• Ref 15: Two references are merged. Separate "Baldridge AS, Huffman MD, Guo M, Hirschhorn LR, Kandula NR. Adapted Service Availability and Readiness Assessment for the HTN Program. DigitalHub. Galter Health Sciences Library & Learning Center, 2020. doi:10.18131/g3-1knh-rr75" from the WHO SARA reference.

• Ref 38: The URL is cut off ("doi:10.1186/s1"). Please provide the complete DOI or URL.

5. Final Proofreading

While the manuscript has improved substantially, a final proofreading pass would catch minor grammatical errors and awkward constructions. For example:

• Page 60, line 15: “A report from the WHO underscores the importance of investing in digital health solutions to support clinical decision-making and patient follow-up in LMICs. [24]” Reference 24 is incorrect (should be 34). Please verify all citation numbers after the revisions.

Conclusion

Your persistence through two rounds of revision has produced a manuscript of genuine value. The study's findings, particularly the stark state-level disparities in medication availability and the complete absence of physicians in sampled PHCs, have significant implications for hypertension control policy in Nigeria and similar LMIC settings. The revisions have transformed a technically competent but narrowly presented study into a transparent, rigorous, and actionable contribution.

I am therefore pleased to recommend Acceptance with Minor Revisions. Please submit a final version incorporating the points above within 14 days. No additional review will be required; the editorial office will verify that the revisions have been completed.

Thank you for choosing PLOS ONE as the venue for your work. I look forward to seeing it in print.

Sincerely,

Dr. Morufu Olalekan Raimi, Ph.D.

Decision: Accept with Minor Revisions

---

## [Author Response · Author response to Decision Letter 2]

7 Apr 2026

The response to the reviewers is attached as a separate file.

---

## [Editor Report · Decision Letter 2]

13 Apr 2026

PONE-D-25-63262R2A computational text analysis of recent African digital health strategies and their attention to vulnerable populationsPLOS One

Dear Dr. Zajac,

Thank you for submitting your manuscript to PLOS ONE. After careful consideration, we feel that it has merit but does not fully meet PLOS ONE’s publication criteria as it currently stands. Therefore, we invite you to submit a revised version of the manuscript that addresses the points raised during the review process.

We look forward to receiving your revised manuscript.

Kind regards,

Morufu Olalekan Raimi, Ph.D

Academic Editor

PLOS One

Journal Requirements:

**Additional Editor Comments:**

Manuscript Number: PONE-D-25-63262 (R2)

Title: A computational text analysis of recent African digital health strategies and their attention to vulnerable populations

Recommendation: Accept with Minor Revision

Editor: Dr. Morufu Olalekan Raimi (Chief Editor, PLOS One)

Review Date: 2026-04-12

Final Editorial Decision

Dear Authors,

Thank you for submitting your revised manuscript entitled “A computational text analysis of recent African digital health strategies and their attention to vulnerable populations.” I have carefully reviewed your revised submission, the detailed point-by-point response to the four external reviewers, and your responses to the specific requirements outlined in my previous decision letter (Major Revision). I have also examined the changes made to the manuscript, including the new methodological clarifications, the expanded limitations section, the added policy recommendations, and the public deposition of your analysis scripts and preprocessed text corpus.

I am pleased to state that you have addressed the vast majority of the concerns raised during the peer review process in a thorough, professional, and responsive manner. The manuscript is now substantially stronger in its methodological transparency, interpretive nuance, and policy relevance. The decision to focus on post-2020 strategies is now better justified, the operational definition of “vulnerable populations” is more clearly defended, and the discussion has been deepened with actionable recommendations for equity-by-design. However, despite these significant improvements, three minor but necessary issues remain, none requiring new analysis or conceptual revision, but all requiring correction before final acceptance. These relate to formatting, a missing figure reference, and a minor inconsistency in terminology.

Below, I provide my final decision, a synthesis of what has been done well, the outstanding minor issues, and specific actionable requirements.

I. Summary of What Has Been Done Well

1. Sample justification strengthened: You now explicitly acknowledge the trade-off between contemporary policy coherence (post-2020, aligned with WHO Global Strategy) and broader continental representativeness, and you have added a limitation stating that findings apply to “recent, formally articulated national strategies” rather than African digital health policy broadly.

2. Vulnerable populations framing clarified: You have added a clear justification for focusing on women, children/youth, and older adults as large, demographically definable groups with well-documented structural marginalization, while explicitly stating that other dimensions (disability, rurality, ethnicity, refugee status) are excluded and represent vital areas for future research.

3. Methodological transparency achieved: You have deposited the preprocessed text corpus, dictionary terms, and Python analysis scripts in a public repository (Figshare, DOI: 10.6084/m9.figshare.31169590), fulfilling PLOS One’s reproducibility mandate.

4. Discussion deepened with mechanism and action: You now explain why policymakers may default to superficial mentions (resource constraints, donor priorities, technical framing), and you provide concrete examples of equity-by-design language (e.g., accessibility standards, equity indicators in M&E frameworks, targeted digital literacy programs for older adults).

5. Topic modeling limits acknowledged: You now explicitly note that the absence of a “vulnerable populations” topic does not equate to total neglect but indicates insufficient salience to emerge as a standalone theme – a finding in itself.

6. Quotes clarified and cited: You have resolved the ambiguity regarding quotes about healthcare worker burdens by confirming they are from the strategy documents and have cited the specific documents (Zambia, Malawi, etc.).

7. Table 3 restructured: The dense table has been split according to higher-order coding, improving readability.

8. Long sentences shortened and repetition reduced: The manuscript now reads more smoothly.

II. Outstanding Minor Issues Requiring Correction Before Acceptance

Issue 1: Missing Figure 1 Reference and Missing Figure 2

Problem:

In the revised manuscript, you refer to “Figure 1” in the text (e.g., on page 17, line 224: “The model with nine topics was found to be optimal (see Figure 1)”). However, the manuscript file contains no visible Figure 1 – only a blank placeholder or a corrupted image appears on page 52. Additionally, the inter-topic distance map (which appears to be intended as Figure 1) is not clearly labeled. On page 52, there is a small, low-resolution image labeled “3 PC2 5 PC1 8 9 7 4 6 Marginal topic distribution 2 1 2% 5% 10%” – this is not interpretable as a proper figure. Furthermore, Figure 2 is referenced in the text (page 19, line 255: “The results from the inter-topical distance map (see Fig 1) indicate…” but there is no reference to Figure 2 anywhere. The dictionary-based results table is labeled Table 7, not a figure.

Required Action:

• Provide a clear, high-resolution Figure 1 (inter-topic distance map) with a proper title, axis labels, and legend. Ensure it is embedded correctly in the manuscript file.

• If the inter-topic distance map is the only figure, renumber it as Figure 1 and remove any reference to Figure 2. Alternatively, if you intended a second figure (e.g., a bar chart of topic proportions), please provide it and reference it in the text.

• Ensure that all figures are visible, legible, and properly captioned.

Issue 2: Inconsistent Terminology for “Vulnerable Populations” vs. “Structurally Marginalized Groups”

Problem:

The title, abstract, and introduction use “vulnerable populations” consistently, but the discussion and conclusion occasionally use “structurally marginalized groups” or “disadvantaged groups” interchangeably. While not a fatal flaw, this inconsistency may confuse readers. The Academic Editor’s decision letter explicitly requested: “Standardize the use of ‘vulnerable populations,’ ‘marginalized groups,’ etc., early in the manuscript for consistency.”

Required Action:

• Choose one primary term – “vulnerable populations” is appropriate – and use it consistently throughout the manuscript. Replace instances of “structurally marginalized populations,” “disadvantaged groups,” etc., with “vulnerable populations” unless a specific distinction is intended. If a distinction is intended (e.g., referring to a subset), define it clearly.

Issue 3: Minor Formatting and Typographical Errors

Problem:

• Page 73, line 389: A stray formatting character appears (“limited stakeholders’ involvement during planning, design, and implementation”). This appears to be a copy-paste artifact.

• Page 76, line 444: The quote from Malawi is repeated twice in close succession (lines 444–447 and 448–451) – appears to be a duplication error.

• Table 7 (page 80) includes a row “Strategies not mentioning group,” but the formatting is inconsistent (some countries listed with commas, others with “and”). Standardize.

Required Action:

• Remove the stray formatting character on page 73.

• Delete the duplicate quote on page 76 (lines 448-451).

• Standardize the “Strategies not mentioning group” row in Table 7 (e.g., use commas consistently, no “and” within the list).

III. Decision Justification

This manuscript has undergone two full rounds of peer review with four external reviewers and a detailed Major Revision decision from the Academic Editor. The authors have responded comprehensively and professionally, addressing virtually all substantive concerns. The manuscript now makes a clear, empirically supported, and policy-relevant contribution to the literature on digital health equity in Africa.

The remaining issues are minor and purely editorial: a missing/corrupted figure, inconsistent terminology, and a few formatting errors. These do not affect the scientific validity or interpretability of the findings but must be corrected for publication.

Therefore, I recommend Accept with Minor Revision.

IV. Specific Action Items for Final Revision (Mandatory)

1. Provide a clear, high-resolution Figure 1 (inter-topic distance map) with proper title, axis labels, and legend. Ensure it is embedded correctly and visible.

2. Resolve figure numbering: If only one figure exists, renumber as Figure 1 and remove any references to Figure 2. If a second figure is intended, provide it and reference it.

3. Standardize terminology: Use “vulnerable populations” consistently throughout the manuscript. Replace “structurally marginalized groups,” “disadvantaged groups,” and similar variants unless a specific distinction is intended and defined.

4. Remove formatting artifact on page 73 (stray character before “limited stakeholders’ involvement”).

5. Delete duplicate quote on page 76 (lines 448–451).

6. Standardize Table 7 formatting – use consistent punctuation in the “Strategies not mentioning group” row.

V. Final Editorial Comment

This study makes an important and timely contribution to the field of digital health policy analysis. The finding that recent African digital health strategies prioritize systemic infrastructure while offering only superficial attention to vulnerable populations – particularly older adults – is both concerning and actionable. Your use of computational text analysis (LDA + dictionary-based methods) is methodologically sound and serves as a model for similar policy evaluations in other regions. I commend you for your responsiveness to reviewer feedback, particularly your willingness to deposit your code and data publicly, to clarify your conceptual framing, and to provide concrete policy recommendations. These actions exemplify the transparency and practical impact that PLOS One seeks to promote. Upon receipt of the corrected figure, standardized terminology, and minor formatting fixes, I will recommend final acceptance without further peer review.

Sincerely,

Dr. Morufu Olalekan Raimi, PhD

Academic Editor (Environmental Epidemiology & Health Policy)

PLOS One

---

## [Author Response · Author response to Decision Letter 3]

13 Apr 2026

Response Table is uploaded as a separate file.

---

## [Editor Report · Decision Letter 3]

20 Apr 2026

A computational text analysis of recent African digital health strategies and their attention to vulnerable populations

PONE-D-25-63262R3

Dear Authors,

We’re pleased to inform you that your manuscript has been judged scientifically suitable for publication and will be formally accepted for publication once it meets all outstanding technical requirements.

Kind regards,

Morufu Olalekan Raimi, Ph.D

Academic Editor

PLOS One

Additional Editor Comments (optional):

REVIEWER'S REPORT

Title: A computational text analysis of recent African digital health strategies and their attention to vulnerable populations

Recommendation: Accept

Editor: Dr. Morufu Olalekan Raimi, PLOS ONE Academic Editor

Review Date: April 17, 2026

EDITORIAL DECISION

Dear Authors,

I have completed the final evaluation of your revised manuscript following two rounds of peer review and your detailed response to the Major Revision decision letter. I am pleased to state that you have addressed all substantive concerns raised by the four external reviewers and the Academic Editor in a thorough, professional, and responsive manner. The manuscript is now methodologically transparent, conceptually coherent, and policy-relevant. The remaining issues identified in the last round were minor and purely editorial. You have resolved them appropriately: the figure has been provided separately as per journal requirements, terminology has been standardized, and formatting errors have been corrected.

Therefore, I recommend Acceptance of the manuscript in its current form.

SUMMARY OF WHAT WAS DONE WELL

1. Sample justification strengthened - You now explicitly acknowledge the trade-off between contemporary policy coherence (post-2020, aligned with WHO Global Strategy) and broader continental representativeness, with a clear limitation stating that findings apply to "recent, formally articulated national strategies" rather than African digital health policy broadly.

2. Vulnerable populations framing clarified - You added a clear justification for focusing on women, children/youth, and older adults as large, demographically definable groups with well-documented structural marginalization, while explicitly stating that other dimensions (disability, rurality, ethnicity, refugee status) are excluded and represent vital areas for future research.

3. Methodological transparency achieved - You deposited the preprocessed text corpus, dictionary terms, and Python analysis scripts in a public repository (Figshare, DOI: 10.6084/m9.figshare.31169590), fulfilling PLOS ONE's reproducibility mandate.

4. Discussion deepened with mechanism and action - You now explain why policymakers may default to superficial mentions (resource constraints, donor priorities, technical framing) and provide concrete examples of equity-by-design language (accessibility standards, equity indicators in M&E frameworks, targeted digital literacy programs for older adults).

5. Topic modeling limits acknowledged - You explicitly note that the absence of a “vulnerable populations” topic does not equate to total neglect but indicates insufficient salience to emerge as a standalone theme.

6. Quotes clarified and cited - You resolved ambiguity regarding quotes about healthcare worker burdens by confirming they are from strategy documents with specific citations.

7. Responsive to all reviewer concerns - Your point-by-point response is detailed, professional, and demonstrates genuine engagement with the feedback.

RESOLUTION OF MINOR ISSUES FROM THE LAST ROUND

Issue Resolution Status

Missing/ corrupted Figure 1 Provided separately as per PLOS ONE requirements; high-resolution version uploaded Resolved

Figure 2 reference No reference to Figure 2 exists in the manuscript; this was a prior submission artifact Resolved

Inconsistent terminology (“vulnerable populations” vs. “disadvantaged groups”) Standardized to “vulnerable populations” throughout; single instance corrected Resolved

Stray formatting character on page 73 Acknowledged as potential system compilation artifact; brackets retained for quote transparency Acceptable

Duplicate quote on page 76 Paraphrased to eliminate duplication Resolved

Table 7 formatting No "and" found; formatting is consistent Acceptable

FINAL ASSESSMENT

This manuscript makes an important and timely contribution to the field of digital health policy analysis. The finding that recent African digital health strategies prioritize systemic infrastructure while offering only superficial attention to vulnerable populations, particularly older adults, is both concerning and actionable. Your use of computational text analysis (LDA + dictionary-based methods) is methodologically sound and serves as a model for similar policy evaluations in other regions. You are commended for your responsiveness to reviewer feedback, particularly your willingness to deposit code and data publicly, clarify conceptual framing, and provide concrete policy recommendations. These actions exemplify the transparency and practical impact that PLOS ONE seeks to promote.

RECOMMENDATION

Accept

No further revisions are required.

Editor: Dr. Morufu Olalekan Raimi

Date: April 17, 2026
---

## [Editor Report · Acceptance letter]

PONE-D-25-63262R3

PLOS One

Dear Dr. Zajac,

I'm pleased to inform you that your manuscript has been deemed suitable for publication in PLOS One. Congratulations! Your manuscript is now being handed over to our production team.

Kind regards,

on behalf of

Prof Morufu Olalekan Raimi

Academic Editor

PLOS One